# Baseline and Kinetic Circulating Tumor Cell Counts Are Prognostic Factors in a Prospective Study of Metastatic Colorectal Cancer

**DOI:** 10.3390/diagnostics11030502

**Published:** 2021-03-12

**Authors:** Virgílio Souza e Silva, Emne Ali Abdallah, Angelo Borsarelli Carvalho de Brito, Alexcia Camila Braun, Milena Shizue Tariki, Celso Abdon Lopes de Mello, Vinicius Fernando Calsavara, Rachel Riechelmann, Ludmilla Thomé Domingos Chinen

**Affiliations:** 1Department of Medical Oncology, A.C. Camargo Cancer Center, São Paulo 01509-900, Brazil; virgilio.silva@accamargo.org.br (V.S.eS.); angelo.brito@accamargo.org.br (A.B.C.d.B.); milena.tariki@accamargo.org.br (M.S.T.); celso.mello@accamargo.org.br (C.A.L.d.M.); rachel.riechelmann@accamargo.org.br (R.R.); 2International Research Center, A.C. Camargo Cancer Center, São Paulo 01508-010, Brazil; emne.abdallah@accamargo.org.br (E.A.A.); alexciabraun@gmail.com (A.C.B.); vinicius.calsavara@accamargo.org.br (V.F.C.)

**Keywords:** circulating tumor cells, kinetics, metastatic colorectal cancer, prognosis, multidrug resistance protein 1 (MRP-1)

## Abstract

The discovery of predictive biomarkers in metastatic colorectal cancer (mCRC) is essential to improve clinical outcomes. Recent data suggest a potential role of circulating tumor cells (CTCs) as prognostic indicators. We conducted a follow-on analysis from a prospective study of consecutive patients with mCRC. CTC analysis was conducted at two timepoints: baseline (CTC1; before starting chemotherapy), and two months after starting treatment (CTC2). CTC isolation/quantification were completed by ISET^®^ (Rarecells, France). CTC expressions of drug resistance-associated proteins were evaluated. Progression-free survival (PFS) and overall survival (OS) were estimated by the Kaplan–Meier method. Seventy-five patients were enrolled from May 2012 to May 2014. A CTC1 cut-off of >1.5 CTCs/mL was associated with an inferior median OS compared to lower values. A difference of CTC2−CTC1 > 5.5 CTCs/mL was associated with a reduced median PFS. By multivariate analysis, CTC1 > 1.5 CTCs/mL was an independent prognostic factor for worse OS. Multi-drug resistance protein-1 (MRP-1) expression was associated with poor median OS. CTC baseline counts, kinetics, and MRP-1 expression were predictive of clinical outcomes. Larger studies are warranted to explore the potential clinical benefit of treating mCRC patients with targeted therapeutic regimens guided by CTC findings.

## 1. Introduction

Colorectal cancer (CRC) is one of the most common gastrointestinal tumors, constituting the second most common cancer diagnosed in women and third most common in men, and accounting for approximately 10% of all incident cancers and cancer-related deaths worldwide [1]. Approximately 50% to 60% of CRC patients develop metastases, with 80% to 90% of these patients afflicted with unresectable hepatic lesions [2,3,4].

The management of metastatic colorectal cancer (mCRC) has advanced significantly in recent years [5]. Treatments have evolved from the time when 5-fluorouracil (5-FU) was the only active agent and conferred an overall survival (OS) of approximately 11 to 12 months, to current therapies that result in an average OS that is reaching three years. These improvements have been driven primarily by the availability of new active agents that include cytotoxic drugs such as oxaliplatin and irinotecan, and monoclonal antibodies that inhibit angiogenesis and proliferation pathways by targeting vascular endothelial growth factor and the epidermal growth factor receptor, respectively [5,6,7,8,9].

Despite these advances, the heterogeneity of mCRC impedes efforts by the oncologist to design targeted treatment strategies and to optimize personalized therapy. Consequently, the development of predictive tools that can be applied to each patient before the initiation of treatment becomes essential for clinical practice [10,11,12]. In addition, the discovery of circulating biomarkers to improve patient management is imperative due to the limitations of the currently available methods of tissue biopsy and radiological evaluation for follow-up and prognosis. Thus, in the era of precision medicine, the liquid biopsy has become an important decision-support tool for the oncologist in the design of treatment strategies [13,14,15,16,17].

Liquid biopsy constitutes real time analysis of tumor and/or metastasis components by body fluids, such as blood, urine, feces, saliva and allows the assessment of the complexity and heterogeneity of the disease. Liquid biopsy components are circulating tumor cells, circulating tumor DNA and microRNAs, tumor-derived extracellular vesicles and thus, provides a less invasive and dynamic way to predict a recurrence and resistance to the proposed treatment [18]. Circulating tumor cells (CTCs) are released by primary and/or metastatic cancer during their formation and/or tumor progression [19]. Numerous previous studies have shown that CTC counts can predict progression-free survival (PFS) and OS in patients with early and metastatic breast cancer, metastatic prostate cancer and CRC [20,21,22,23]. There are currently 963 clinical trials using CTCs worldwide (clinicaltrials.gov, accessed on the 9 March 2021), representing the interest in the applicability of liquid biopsy in the management of oncology patients [24].

Therefore, there are perspectives for CTCs to be used in the CRC for screening (early detection of invasive cancers); in localized cancer (risk stratification, prognosis and monitoring after treatment); and in metastatic disease (treatment design, response monitoring and identification of drug resistance mechanisms). However, clinical validation is still necessary [25,26,27]. In addition, as there are many different methods to isolate CTCs, it is also fundamental to standardize the methodology and many international efforts have been made in this sense. The majority of the technologies described in the literature use EpCAM to isolate these cells, however, there are controversies about the use of this molecule to isolate CTCs from colorectal tumor origins. In the review paper from Eslami-S et al. [28], the authors comment about the variation of EpCAM positive CTCs among different types of solid cancers, that high numbers of EpCAM positive CTCs are often detected in blood samples from patients with breast, prostate and small cell lung cancer. Those counts (EpCAM + CTC) are low in patients with pancreatic, colorectal and non-small cell lung cancer. In metastatic colorectal cancer, per example, CTC counts vary greatly between studies, with around 10% to 30% of CTC-positive patients at baseline before treatment. Our group have been working with ISET (Isolation by SizE of Tumors, Rarecells, France) for a long time and have published studies in colorectal cancer, with higher detection rates, around 88.5% [29,30]. It can be explained by the possibility to isolate cells independently of the marker, so, CTCs under epithelial–mesenchymal transition can be easily taken.

Here, we have a follow-on study, with the objective to evaluate the real impact of CTCs in OS and PFS. With part of the cohort included, we previously demonstrated the importance of kinetic evaluations of CTCs [26] and assessments of CTC expressions of thymidylate synthase (TYMS) and multidrug resistance protein-1 (MRP-1) as potential predictive biomarkers in mCRC patients [29,31]. So, we analyzed data set published previously by our group (with 54 patients) combined with an additional sample (more 21 patients) to determine if CTC (counts and kinetics and biomarkers) would remain as prognostic indicators.

## 2. Materials and Methods

### 2.1. Patient Recruitment

This follow-on analysis of a prospective longitudinal study was conducted at the A.C. Camargo Cancer Center (Rua Professor Antônio Prudente, 211, Liberdade; São Paulo, Brazil) and was approved by the local Ethics Committee (CEP 1367/10B-10 April 2012). Written informed consent was obtained from all patients prior to enrollment. Inclusion criteria included a diagnosis of inoperable metastatic colorectal cancer, confirmed by histopathology of the primary and/or metastatic lesion and radiological analysis; age of ≥18 years; Eastern Clinical Oncology Group Performance Status (ECOG PS) of 0–2 without organ dysfunction; and an intent to begin chemotherapy for metastatic disease. Cancer stage was determined by the results of a physical examination and diagnostic imaging, and was scored by the Response Evaluation Criteria in Solid Tumors (RECIST) criteria v.1.1 [32] Patients were designated by numerical code to preserve confidentiality.

### 2.2. Study Design

We conducted a prospective cohort study. All participants had 8 mL of blood collected at two different timepoints: baseline (CTC1; before starting treatment with either first, second, or third-line chemotherapy; after the diagnosis of metastasis or tumor progression; or at the initiation of a new chemotherapy regimen); and two months after the start of treatment (CTC2), when radiological imaging was performed.

Patients underwent computed tomography and/or magnetic resonance imaging of the thorax, abdomen, and pelvis before CTC1 collection and approximately every 2–4 months during treatment, or as clinically indicated in the judgment of the attending physician. Images were interpreted using the RECIST criteria v.1.1. The clinicopathological information of the patients and the mutational status of *RAS* in the tumors were collected from medical records.

### 2.3. CTC Isolation and Identification

Blood (8.0 mL) was collected in ethylenediaminetetraacetic acid (EDTA) tubes and maintained under homogenization until processing (approximately 4 h) by the Isolation by SizE of Tumor cell technique (ISET^®^; Rarecells, France) according to the manufacturer’s instructions. To identify and analyze CTCs, we performed immunocytochemistry (ICC) using the following antibodies: anti-thymidilate synthase (TYMS) (WH0007298M1; Sigma-Aldrich, San Luis, MO, USA) to verify 5-FU resistance; anti-excision repair cross-complementation group 1 protein (ERCC1) (SAB4500795; Sigma-Aldrich, San Luis, MO, USA) to verify oxaliplatin resistance; and anti-multidrug resistance-associated protein-1 (MRP-1) (HPA002380; Sigma-Aldrich, San Luis, MO, USA) to determine irinotecan resistance. We also performed ICC against the leukocyte common antigen CD45 (CSB-PA010546; CusaBio, Houston, TX, USA) to exclude leucocytes from our analysis. After the ICC assay (performed as described by Abdallah et al. 2016 [29]), spots on ISET^®^ membranes were counterstained with hematoxylin and analyzed by light microscopy. We conducted cytopathologic analysis of CTCs according to the following parameters: high nuclear–cytoplasmic ratio (>0.8), cell diameter >16 µm; hyperchromatic and irregular nuclei [33]; and negative CD45 reactivity.

### 2.4. CTC Count

We considered the CTC count as a continuous variable and calculated cut-off points to discriminate between two groups of patients, those with good versus those with poor clinical outcomes. The cut-offs for each event of interest (OS and PFS) were estimated as previously reported [34].

### 2.5. Statistical Analysis

Baseline patient characteristics were expressed as absolute and relative frequencies for qualitative variables and as the median, minimum and maximum for quantitative variables. Regarding CTC counts at the two timepoints, the determination of two groups of observations by a simple cut-off point was estimated using the maximum of the standardized log-rank statistic [34]. Simple cut-off points to differentiate groups of observations CTC counts at each timepoint were estimated by using the maximum of the standardized log-rank statistic. In other words, CTC1 and delta CTC (CTC2-CTC1) values were calculated by Lausen and Schumacher method [34], which aims to determine the best cut-off point in order to discriminate the survival functions. The value 1.5 for CTC1 and 5.5 for delta CTC obtained were placed as cut-offs, as qualitative variables (Appendix A). Then, we estimated the survival curves using the Kaplan–Meier estimator and the covariate’s effect was evaluated by means of Cox regression proportional hazards model.

Survival functions were estimated by the Kaplan–Meier method, and curves were compared by using the log-rank test. The Cox semi-parametric proportional hazards model was fitted to evaluate relationships between covariates and overall survival/progression-free survival [35]. The assumption of proportional hazards was assessed on the so-called Schoenfeld residuals [36,37]. There was evidence that covariates had a constant effect over time in all cases. Overall survival was calculated from the date of CTC1 collection to the date of first recurrence, determined by diagnostic imaging or last follow-up. The significance level was fixed at 5% for all tests. Statistical analyses were performed using IBM SPSS Statistics version 23.0 (IBM Corp., Armonk, NY, USA) and R software version 3.5 (R Foundation for Statistical Computing, Vienna, Austria).

## 3. Results

### 3.1. Patient Characteristics

Seventy-five consecutive mCRC patients were enrolled from May 2012 to May 2014. Mean age was 59 years (24–81 years), and most were male (*n* = 42; 56%). All had metastatic disease at the time of enrollment, including 22.7% (*n* = 17) with metastasis confined to the liver, and 37.3% (*n* = 28) with extra-hepatic and hepatic metastasis. Most patients had an ECOG PS of 0 or 1, and none had contraindications to systemic chemotherapy. Forty-one percent (*n* = 31) had undergone previous surgical resections of metastases; 25% (*n* = 19), 12% (*n* = 9), and 4% (*n* = 3) had resections of hepatic, pulmonary, and hepatic and pulmonary metastases, respectively (Table 1). Baseline CTC1 analysis was completed before the initiation of first-line chemotherapy in 38/75 patients (50.6%); before the initiation of second-line therapy (after disease progression during first-line therapy) in 20/75 patients (26.7%); and after progression during second-line therapy in 17/75 patients (22.7%).

Median survival was 34.5 months (95% confidence interval (CI), 28.3–40.7 months). *RAS* mutants were associated with a median survival of 20.1 months (95% CI, 10.7–29.5 months), which was significantly shorter than that of patients with wild-type *RAS*, in whom median survival was 41.0 months (95% CI, 32.3–49.7 months); log-rank *p* = 0.001). Patients with right-sided colon tumors had an inferior median survival (20.5 months, 95% CI, 11.3–29.9) when compared with patients with left-sided cancers (39.5, 95% CI, 30.3–48.7 months) (log-rank *p* = 0.015).

### 3.2. Prognostic Value of CTCs

The best estimated cut-off point for CTC1 and OS for the entire cohort was 1.5 CTCs/mL (median 2.5 CTCs/mL, range 0–31.2 CTCs/mL). Patients with CTC1 > 1.5 had a median OS of 24.5 months (95% CI, 9.5–39.4 months), less than the OS of patients with CTC ≤ 1.5 (34.2 months (95% CI, 18.4–50.2 months); log-rank *p* = 0.041) (Figure 1).

Patients with CTC1 > 1.5 had a higher risk of death than patients with CTC1 ≤ 1.5 CTCs/mL (hazard ratio (HR) = 1.893; 95% CI, 1.015–3.528; *p* = 0.041) (Table 2). Patients with CTC2—CTC1 > 5.5 per mL demonstrated poorer median progression-free survival (PFS) (3.2 months; 95% CI, 0.001–6.5 months) when compared to CTC2—CTC1 ≤ 5.5 (9.1 months; 95% CI, 7.1–11.1 months; log-rank *p* = 0.005). Patients with CTC2–CTC1 > 5.5 had a higher risk of death than those with CTC2–CTC1 ≤ 5.5 CTCs/mL (HR = 3.107; 95% CI, 1.34–7.22; *p* = 0.01) (Table 3).

Among the 38 patients receiving first-line therapy, those with CTC1 > 1.0 had a median OS of 29.74 months (95% CI, 17.9–41.6); in patients with CTC1 ≤ 1.0 median OS was not reached (log rank *p* = 0.09; HR = 2.5, 95% CI, 0.83–7.53, *p* = 0.103). We estimated a cut-off of CTC2−CTC1 > 2.0/mL for PFS in this group. The median PFS was 4.5 months (95% CI, 1.7–7.3 months) for patients with CTC2−CTC1 > 2.0/mL, and 22.3 months (95% CI, 8.0–36.5 months) for those with CTC2−CTC1 ≤ 2.0/mL (log rank *p* = 0.001).

For the 20 patients undergoing second-line treatment, the estimated CTC1 cut-off was 1.65 CTCs/mL for OS. Median OS values were 29.0 months (95% CI, 29.9–58.0) for patients with CTC1 > 1.65/mL, and 43.5 months (95% CI, 9.2–48.9 months) for those with CTC1 ≤ 1.65/mL (log rank *p* = 0.29). In this group, the estimated cut-off for PFS was CTC2−CTC1 was 1.0 CTCs/mL; patients with CTC2−CTC1 ≤ 1.0 showed a median PFS of 14.0 months (95% CI, 2.3–25.7 months) versus 8.8 months (95% CI, 5.1–12.6 months) for those with >1.0/mL (log rank *p* = 0.36).

### 3.3. CTC Biomarkers

Evaluation of CTC MRP-1 expression was limited to 19 patients due to restricted availability of materials (here, we evaluated the same 19 patients included in the paper published by Abdallah et al. [29] and analyzed the power of the variable MRP-1 with the extended follow-up). MRP-1 expression was associated with worse median OS (4.21 months, 95% CI: 0.001–12.4 months) compared to negative MRP-1 expression (32.6 months, 95% CI, 19.2–46 months; log rank *p* = 0.026). Despite the limited sample size, we demonstrated that MRP-1 expression in CTCs increased the risks of death (HR = 4.791, 95% CI, 1.05–21.83; *p* = 0.043) and disease progression (HR = 6.933, 95% CI, 1.69–28.45; *p* = 0.007) (Table 2 and Table 3).

ERCC1 expression was evaluated in 13 patients (data previously published by Abdallah et al. [29] with updated follow-up). There was a trend towards shorter OS in ERCC1-positive compared to ERCC1-negative patients that did not reach statistical significance (median OS of 29.7 months (95% CI, 10.9–48.5 vs. 36.8, 95% CI, 0.001–102 months; log-rank *p* = 0.38). TYMS expression was analyzed in 36 patients (data previously published by Abdallah et al. [31] with updated follow-up). TYMS-positive patients experienced a median OS of 16.97 months (95% CI, 10.2–23.7) versus 26.8 months (95% CI, 20.6–33.1 months) for TYMS-negative patients (log-rank *p* = 0.55).

### 3.4. Multivariate Analysis

To evaluate the effects of independent variables on OS and PFS, we fitted a multiple Cox regression model for each outcome considering CTC1, *RAS* status, and laterality (right- versus left-sided colon as site of primary tumor) for the OS outcome; and CTC2–CTC1, *RAS* status, and laterality for the PFS outcome. Patients with CTC1 > 1.5 had a higher risk of death than those with CTC1 ≤ 1.5 CTCs/mL (HR = 2.34; 95% CI, 1.113–4.919; *p* = 0.025), and were also more likely to harbor *RAS* mutations (HR = 5.467; 95% CI, 2.32–12.886; *p* = 0.001) (Table 2 and Table 3).

Multivariate analysis disclosed that *RAS* mutations were associated with shorter OS and PFS. Right-sided primary tumors were associated with a trend toward decreased OS and PFS. Unfortunately, MRP-1 expression was not analyzed in the multiple regression model due to the limited sample size (Table 2 and Table 3).

## 4. Discussion

Due to the heterogeneity of mCRC, we have not yet discovered a sensitive and specific prognostic biomarker that can be applied in all cases to design optimal personalized treatment strategies [38,39]. We currently devise therapies for mCRC patients based on symptoms, primary tumor site, previous treatments, cancer stage, molecular evaluation (*BRAF*, *KRAS* or *NRAS* mutations and microsatellite instability), comorbidities, and treatment goals [5,40,41,42]. However, improvement of this initial assessment process is crucial to enhance clinical outcomes. Advances provided by liquid biopsy open new opportunities for the discovery of biomarkers to forecast more accurate prognoses, to assess drug resistance before and during therapy, and to monitor treatment response [11,25,43]. Consequently, the study of CTCs in mCRC is essential, as it allows quantitative and kinetic evaluations, as well as the identification of drug resistance-associated proteins, thus enabling patient-centric prognosis and evaluations of therapeutic efficacy [13,26,44,45]. CTC analysis provides a real-time image of various tumor characteristics, including the extent of heterogeneity at specific timepoints.

Numerous studies have demonstrated that baseline CTC quantification has prognostic relevance. Herein, our study of 75 mCRC patients receiving different lines of chemotherapy and varied treatment regimens showed that CTC1 > 1.5/mL was an independent prognostic indicator of OS. In addition, CTC1 > 1.5 CTCs/mL was associated with decreased OS in patients receiving first-line therapy. We suggest that CTC quantification before the start of chemotherapy, combined with other clinical criteria and molecular diagnostics, can predict clinical outcomes and thereby assist the physician in designing treatment.

In an earlier study of 54 patients, we demonstrated that CTC kinetics were important prognostic indicators [26]. Our current study associated higher increases in CTC counts with reduced PFS in the entire cohort and in first-line chemotherapy recipients (CTC2−CTC1 cut-offs of >5.5 per mL and >2.0/mL, respectively). These results are concordant with our previous findings and demonstrate the clinical relevance of CTC kinetics.

In addition to conducting quantitative and kinetic analyses of CTCs, we assessed drug resistance-associated protein expressions in CTCs that may predict treatment response and enable the selection of personalized treatment regimens. Consequently, this study evaluated proteins investigated by our group in previous publications [29,31]. CTC MRP-1 expression was related to reduced PFS in an earlier study by our group [29]. In the current analysis, we confirmed that in addition to the previously demonstrated relationship with worse PFS, MPR-1 expression was associated with a major reduction in OS. Yang et al. [46] evaluated 116 patients with colon adenocarcinoma, and related MPR-1 expression in tumor tissue to poor prognosis. Their results, together with ours, highlight the clinical relevance of CTC MRP-1 expression as a prognostic biomarker and predictor of therapeutic response that can be used in future clinical studies and, if possible, in clinical practice to select MRP-1-positive patients for more intensive treatments.

## 5. Conclusions

In conclusion, our study of a heterogenous sample of 75 mCRC patients demonstrated the clinical utility of CTC assessments. Quantitative, kinetic, and resistance protein analyses of CTCs, together with other clinical and molecular factors, allow improved individualized prognostic forecasts. Our study suggests that CTCs represent a promising tool for future clinical investigations to corroborate these findings, and thus enable the oncologist to select patients whose cancers are more likely to respond to particular treatment regimens, and to optimize the sequence of therapeutic options for patients with mCRC.

## Figures and Tables

**Figure 1 diagnostics-11-00502-f001:**
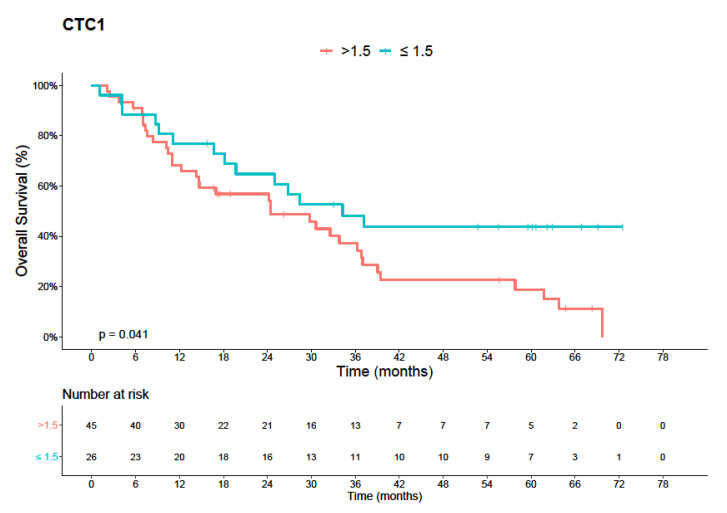
Kaplan–Meier curves of overall survival in the general studied population (75 patients with metastatic colorectal cancer). In blue, survival curve for patent with ≤1.5 CTC/mL. In red, patients with >1.5 CTC/mL (*p* = 0.041).

**Table 1 diagnostics-11-00502-t001:** Clinicopathological features.

Variable	No	%
Total number of patients	75	100
Age (in years)		
Median (Min-Max); Mean (SD)	59 (24–81)	57.32 (12.87)
Gender		
Male	42	56
Female	33	44
ECOG OS		
0	43	60
1	25	33
2	7	7
Laterality		
Left side	54	72
Right side	19	25.3
Unknown	2	2.7
Location of metastasis		
Only hepatic	17	22.7
Hepatic and extra-hepatic	28	37.3
Other (pulmonary, lymph node and or peritoneal)	30	40
First-line treatment		
Oxaliplatin-based	41	54.6
Irinotecan-based	32	42.7
No chemotherapy	2	2.7
Line of treatment (at CTC1 collection)		
First-line	38	50.6
Second-line	20	26.7
Third-line	17	22.7
Metastasis resection		
Hepatic	19	25
Pulmonary	9	12
Hepatic and Pulmonary	3	4
No resection	44	59
*RAS* status		
Wild-type	41	54.7
Mutant *	33	44
Unknown	1	1.3

ECOG PS: Eastern Cooperative Oncology Group Performance Status; SD: standard deviation. * Related to the 33 patients with *RAS* mutations, 21 had *KRAS* codon 12 mutations, nine had *KRAS* codon 13 mutations, two had *KRAS* codon 146 mutations, and one had an *NRAS* codon 12 mutation.

**Table 2 diagnostics-11-00502-t002:** Estimate of the parameters of the simple and multiple Cox regression model for overall survival from metastatic colon cancer.

Variable	Category	Simple Cox Regression Model			Multiple Cox Regression Model		
		HR	95% CI	*p*	HR	95% CI	*p*
**CTC1**	≤1.5	Ref			Ref		
	>1.5	1.893	1.015–3.528	0.041	2.34	1.113–4.919	0.025
**MRP1**	Negative						
	Positive	4.791	1.051–21.829	0.043			
***RAS* status**	Wild	Ref			Ref		
	Mutated	3.306	1.672–6.531	0.001	5.467	2.320–12.886	<0.0001
**Laterality**	Left side	Ref			Ref		
	Right side	2.246	1.178–4.282	0.014	2.166	0.984–4.707	0.055

HR: hazard ratio; Ref: reference category.

**Table 3 diagnostics-11-00502-t003:** Estimate of the parameters of the simple and multiple Cox regression model for progression-free survival from metastatic colon cancer.

Variable	Category	Simple Cox Model			Multiple Cox Model		
		HR	95% CI	*p*	HR	95% CI	*p*
**CTC2−CTC1**	≤5.5	Ref			Ref		
	>5.5	3.107	1.340–7.220	0.010	2.373	0.950–5.918	0.064
**MRP1**	Negative	Ref					
	Positive	6.933	1.689–28.454	0.007			
***RAS* status**	Wild	Ref			Ref		
	Mutated	3.044	1.602–5.783	0.001	8.362	2.719–25.717	<0.0001
**Laterality**	Left side	Ref			Ref		
	Right side	1.659	0.929–2.902	0.087	0.826	0.339–2.009	0.673

HR: hazard ratio; Ref: reference category.

## Data Availability

Not applicable. The data presented in this study are available on request from the corresponding author. The data are not publicly available because they are organized in a bank data that does not allow public sharing.

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
