# Peer review of "Baseline and Kinetic Circulating Tumor Cell Counts Are Prognostic Factors in a Prospective Study of Metastatic Colorectal Cancer"

_diagnostics, 2021, doi:10.3390/diagnostics11030502_

Round 1

Reviewer 1 Report

The authors address a topic of high scientific impact: the circulating tumor cells as tools for liquid biopsy in colorectal cancer. However, the introduction section might benefit from a wider description of the topic and its impact. Regarding the experimental section, the expression of EpCam marker should be included in the study for validating the CTCs in whole blood. The following paper could be useful to improve the current manuscript:

https://doi.org/10.31925/farmacia.2018.5.16

Author Response

Answer:

Dear Reviewer 1, thank you for your time reading and evaluating our paper. We also thank you for your suggestion about the introduction, it improved our study. We insert a paragraph in the introduction section about liquid biopsy, its compartments and clinical  applications, as you can see here:

“Liquid biopsy constitutes real time analysis of tumor and/or metastasis components by body fluids, such as blood, urine, feces, saliva and allows the assessment of the complexity and heterogeneity of the disease. Liquid biopsy components are circulating tumor cells, circulating tumor DNA and microRNAs, tumor derived extracellular vesicles and thus, provides a less invasive and dynamic way to predict a recurrence and resistance to the proposed treatment18. Circulating tumor cells (CTCs) are released by primary and / or metastatic cancer during their formation and / or tumor progression19 .  Numerous previous studies have shown that CTC counts can predict progression-free survival (PFS) and OS in patients with early and metastatic breast cancer, metastatic prostate cancer and CRC20-23. There are currently 791 clinical trials using CTCs worldwide (clinicaltrials.gov), representing the interest in the applicability of liquid biopsy in the management of oncology patients24.”

Concerning your suggestion about the EpCAM marker, we read the paper you suggested about the standardization of flow cytometry for blood samples spiked with HT-29 adenocarcinoma cells, it is really interesting. We will try to use the markers the authors used (MUC, pan-CK and EpCAM) here, in our lab, in our flow cytometry equipment. However, we do not think we will be able to test EpCAM in our samples, as we used all spots from ISET membranes we had disposable. In addition, there are controversies about the use of EpCAM to isolate CTCs from colorectal tumors origin. In the review paper from Eslami-S et al, published last year on Cells (Cells 2020, 9, 1836), there is a paragraph where the authors comment about the variation of EpCAM positive CTCs among different types of solid cancers. They comment that high numbers of EpCAM positive CTCs are often detected in blood samples from patients with breast, prostate and small cell lung cancer. Those counts (EpCAM+ CTC) is low in patients with pancreatic, colorectal and non-small cell lung cancer. In metastatic colorectal cancer, per example, CTC counts vary greatly between studies, with around 10% to 30% of CTC-positive patients at baseline before treatment (Eslami-S et al., 2019). Moreover, our group have been working with ISET for a long time and have publishing studies with colorectal cancer, with higher detection rates, around 88.5% (Buim et al., 2015; Abdallah et al., 2016). It can be explained by the possibility to isolate cells independently of the marker, so, CTCs under epithelial-mesenchymal transition can be easily taken.

References:

Abdallah EA, Fanelli MF, Souza E Silva V, et al. MRP1 expression in CTCs confers resistance to irinotecan-based chemotherapy in metastatic colorectal cancer. Int J Cancer 2016; 139(4): 890-898. doi:10.1002/ijc.30082

Buim MEC, Fanelli MF, Souza VS, Romero J, Abdallah EA et al.Detection of KRAS mutations in circulating tumor cells from patients with metastatic colorectal cancer. Cancer Biology & Therapy 16:9, 1289-95.

Eslami-S, Z.; Cortés-Hernández, L.E.; Alix-Panabières, C. Circulating tumor cells: Moving forward into clinical applications. Precis. Cancer Med. 2019, 3, 4.

Eslami-S, Z.; Cortés-Hernández, L.E.; Alix-Panabières, C. Epithelial Cell Adhesion Molecule: An Anchor to Isolate Clinically Relevant Circulating Tumor Cells. Cells 2020, 9, 1836.

Reviewer 2 Report

The authors aimed to identify a sensitive and specific prognostic biomarker for metastatic colorectal cancer. While they focused on circulating tumor cells, they are not specific to colorectal cancer. Furthermore, metastatic cancer cells after EMT may not be identified by a regular procedure for detecting circulating tumor cells using Ep-CAM, an epithelial-specific marker.

Major

  • Ep-CAM marker: circulating tumor cells are in general isolated using an antibody against Ep-CAM. Ep-CAM is a marker for epithelial cells. Metastatic cancer cells are likely to be experienced EMT (epithelial to mesenchymal transition), and thus it is difficult to collect metastatic tumor cells using a standard procedure. Since the study is targeted to identify a marker for metastatic colorectal cancer, the described approach does not seem to be appropriate.
  • Threshold values: the threshold values for CTC1 and (CTC2-CTC1) are set to 1.5 and 5.5, respectively. However, it is unclear how these values are determined. The authors should show the effect of these values on the p-values.
  • Integrated analysis: the statistical analysis is conducted separately on 4 items as CTC1, MRP1, Ras status, and laterality. The authors should conduct statistical analysis using a set of these items.
  • Use of circulating tumor cells: Circulating tumor cells are generated by many types of cancers and they are not specific to metastatic colorectal cancer.

Minor

  • Figure 1: The quality of this figure is poor because of the distorted aspect ratio. The p-value of 0.041 does not support the entity as a sensitive and specific prognostic biomarker.
  • Abbreviations: it is difficult to read because of too many abbreviated terms. For instance, the sentence, the study of CTCs in mCRC is essential, is not readable.
  • Tables 2&3: There is room to improve the format of these tables to make them easier to understand.

Author Response

The authors aimed to identify a sensitive and specific prognostic biomarker for metastatic colorectal cancer. While they focused on circulating tumor cells, they are not specific to colorectal cancer. Furthermore, metastatic cancer cells after EMT may not be identified by a regular procedure for detecting circulating tumor cells using Ep-CAM, an epithelial-specific marker.

Major

  • Ep-CAM marker: circulating tumor cells are in general isolated using an antibody against Ep-CAM. Ep-CAM is a marker for epithelial cells. Metastatic cancer cells are likely to be experienced EMT (epithelial to mesenchymal transition), and thus it is difficult to collect metastatic tumor cells using a standard procedure. Since the study is targeted to identify a marker for metastatic colorectal cancer, the described approach does not seem to be appropriate.

Answer: Dear Reviewer 2, thank you for reading our paper and for making interesting questions. As we also answered Reviewer 1, there are controversies about the use of EpCAM to isolate CTCs from colorectal tumors origin. In the review paper from Eslami-S et al, published last year on Cells (Cells 2020, 9, 1836), there is a paragraph where the authors comment about the variation of EpCAM positive CTCs among different types of solid cancers. They comment that high numbers of EpCAM positive CTCs are often detected in blood samples from patients with breast, prostate and small cell lung cancer. Those counts (EpCAM+ CTC) is low in patients with pancreatic, colorectal and non-small cell lung cancer. In metastatic colorectal cancer, per example, CTC counts vary greatly between studies, with around 10% to 30% of CTC-positive patients at baseline before treatment (Eslami-S et al., 2019). Moreover, our group have been working with ISET for a long time and have publishing studies with colorectal cancer, with higher detection rates, around 88.5% (Buim et al., 2015; Abdallah et al., 2016). It can be explained by the possibility to isolate cells independently of the marker, so, CTCs under epithelial-mesenchymal transition can be easily taken.

Concerning your point about the identification of a marker for metastatic colorectal cancer, our idea was to verify if CTCs could be a useful one. With our work, we can suggest that efforts can be done, with clinical trials, as here, we showed that CTCs seem to be promising.

  • Threshold values: the threshold values for CTC1 and (CTC2-CTC1) are set to 1.5 and 5.5, respectively. However, it is unclear how these values are determined. The authors should show the effect of these values on the p-values.

Answer: for better understanding, we wrote an additional paragraph on Statistics analysis section, as you can see here:

“Baseline patient characteristics were expressed as absolute and relative frequencies for qualitative variables and as the median, minimum and maximum for quantitative variables.  Regarding CTC counts at the two timepoints, the determination of two groups of observations by a simple cut-off point was estimated using the maximum of the standardized log-rank statistic32. Simple cut-off points to differentiate groups of observations CTC counts at each timepoint were estimated by using the maximum of the standardized log-rank statistic. In other words, CTC1 and delta CTC (CTC-2-CTC1) values were calculated by Lausen & Schumacher method32, which aims to determine the best cut-off point in order to discriminate the survival functions. The value 1.5 for CTC1 and 5.5 for delta CTC obtained were placed as cut-offs, as qualitative variables. Then, we estimated the survival curves using the Kaplan-Meier estimator and the covariate´s effect was evaluated by means of Cox regression proportional hazards model”.

  • Integrated analysis: the statistical analysis is conducted separately on 4 items as CTC1, MRP1, Ras status, and laterality. The authors should conduct statistical analysis using a set of these items.

Answer: we did it, it is the multivariate analysis (tables 2 (for overall survival) and 3 (for progression free-survival)).

Table 2: Estimate of the parameters of the simple and multiple Cox regression model for overall survival from metastatic colon cancer.

Variable

Category

Simple Cox regression model

Multiple Cox regression model

HR

95%CI

p

HR

95%CI

P

CTC1

<1.5

Ref

Ref

>1.5

1.893

1.015-3.528

0.041

2.34

1.113-4.919

0.025

MRP1

Negative

Positive

4.791

1.051-21.829

0.043

RAS status

Wild

Ref

Ref

Mutated

3.306

1.672-6.531

0.001

5.467

2.320-12.886

<0.0001

Laterality

Left side

Ref

Ref

Right side

2.246

1.178-4.282

0.014

2.166

0.984-4.707

0.055

HR: hazard ratio; Ref: reference category.

Table 3: Estimate of the parameters of the simple and multiple Cox regression model for progression free-survival from metastatic colon cancer.

Variable

Category

Simple Cox model

Multiple Cox model

HR

95%CI

p

HR

95%CI

p

CTC2-CTC1

<5.5

Ref

Ref

>5.5

3.107

1.340-7.220

0.010

2.373

0.950-5.918

0.064

MRP1

Negative

Ref

Positive

6.933

1.689-28.454

0.007

RAS status

Wild

Ref

Ref

Mutated

3.044

1.602-5.783

0.001

8.362

2.719-25.717

<0.0001

Laterality

Left side

Ref

Ref

Right side

1.659

0.929-2.902

0.087

0.826

0.339-2.009

0.673

HR: hazard ratio; Ref: reference category.

  • Use of circulating tumor cells: Circulating tumor cells are generated by many types of cancers and they are not specific to metastatic colorectal cancer.

Answer: CTCs are released from primary tumors and/or metastatic tumors by the majority of cancers, that´s why they are called “liquid biopsy”. CTCs, as one of the compartments of liquid biopsy, have been proposed as a manner to follow treatment or even, to screen cancer for many scientists (for more information, please, see reviews from Klaus Pantel and Catherine Alix-Panabiere; we also improved our introduction section, according to suggestion of reviewer 1). There are 791 clinical trials (https://clinicaltrials.gov/) using CTCs worldwide, 387 in the United States and 11 in Brazil. As the follow-up of patients with metastatic colorectal cancer is still a challenge, CTCs can be a useful tool to help clinicians to manage the response to treatment.

Minor

  • Figure 1: The quality of this figure is poor because of the distorted aspect ratio. The p-value of 0.041 does not support the entity as a sensitive and specific prognostic biomarker.

Answer: Dear Reviewer 2, of course, a larger study to validate the cut-off point must be done. However, it is important to emphasize that in our cohort there is evidence of a significant effect of CTC in the failure rate (simple and multivariable Cox models), which shows the importance to better investigate CTCs in larger clinical trials.

  Abbreviations: it is difficult to read because of too many abbreviated terms. For instance, the sentence, the study of CTCs in mCRC is essential, is not readable.

Answer: thank you for pointing this, we made the correction.

  • Tables 2&3: There is room to improve the format of these tables to make them easier to understand

Answer: thank you for pointing this, we re-wrote the tables. We hope you appreciate them.

Round 2

Reviewer 1 Report

The authors improved the manuscript and I think it is now good for publication. However, I think it would worth citing the previous suggested work as other groups approached the topic in a different manner and that mention would widen the state of the art section.

Author Response

Dear Reviewer 1, thank you again for your important suggestions, in an intention to improve our paper. We made the changes, as you can see here, in the Introduction section:

“Therefore, there are perspectives for CTCs to be used in the CRC for screening (early detection of invasive cancers); in localized cancer (risk stratification, prognosis and monitoring after treatment); and in metastatic disease (treatment design, response monitoring and identification of drug resistance mechanisms). However, clinical validation is still necessary [25-27] . In addition, as there are many different methods to isolate CTCs, it is also fundamental to standardize the methodology and many international efforts have been made in this sense. The majority of the technologies described in the literature use EpCAM to isolate these cells, however, there are controversies about the use of this molecule to isolate CTCs from colorectal tumors origin. In the review paper from Eslami-S et al. {28}, the authors comment about the variation of EpCAM positive CTCs among different types of solid cancers, that high numbers of EpCAM positive CTCs are often detected in blood samples from patients with breast, prostate and small cell lung cancer. Those counts (EpCAM+ CTC) is low in patients with pancreatic, colorectal and non-small cell lung cancer. In metastatic colorectal cancer, per example, CTC counts vary greatly between studies, with around 10% to 30% of CTC-positive patients at baseline before treatment. Our group have been working with ISET (Isolation by SizE of Tumors, Rarecells, France) for a long time and have publishing studies with colorectal cancer, with higher detection rates, around 88.5% [29,30]. It can be explained by the possibility to isolate cells independently of the marker, so, CTCs under epithelial-mesenchymal transition can be easily taken”

Reviewer 2 Report

In the review of the original manuscripts, 4 major concerns were pointed out. The responses of the authors are not sufficient. In the revised manuscript, there is no description on the evaluation of Ep-CAM. In the revision, there is no attempt to evaluate the effects of the threshold values. The description of the integrated analysis is not sufficient. No clear rationale for using circulating tumor cells for metastatic colorectal cancer. 

Author Response

Reviewer 2

In the review of the original manuscripts, 4 major concerns were pointed out. The responses of the authors are not sufficient. In the revised manuscript, there is no description on the evaluation of Ep-CAM.

Answer: Dear Reviewer 2, as we told in the first review, we will not able to test EpCAM at this time, as we used all ISET membranes. We inserted a paragraph about the use of EpCAM for colorectal cancer, explaining the advantage of ISE, as you can see here:

“Therefore, there are perspectives for CTCs to be used in the CRC for screening (early detection of invasive cancers); in localized cancer (risk stratification, prognosis and monitoring after treatment); and in metastatic disease (treatment design, response monitoring and identification of drug resistance mechanisms). However, clinical validation is still necessary [25-27] . In addition, as there are many different methods to isolate CTCs, it is also fundamental to standardize the methodology and many international efforts have been made in this sense. The majority of the technologies described in the literature use EpCAM to isolate these cells, however, there are controversies about the use of this molecule to isolate CTCs from colorectal tumors origin. In the review paper from Eslami-S et al. {28}, the authors comment about the variation of EpCAM positive CTCs among different types of solid cancers, that high numbers of EpCAM positive CTCs are often detected in blood samples from patients with breast, prostate and small cell lung cancer. Those counts (EpCAM+ CTC) is low in patients with pancreatic, colorectal and non-small cell lung cancer. In metastatic colorectal cancer, per example, CTC counts vary greatly between studies, with around 10% to 30% of CTC-positive patients at baseline before treatment. Our group have been working with ISET (Isolation by SizE of Tumors, Rarecells, France) for a long time and have publishing studies with colorectal cancer, with higher detection rates, around 88.5% [29,30]. It can be explained by the possibility to isolate cells independently of the marker, so, CTCs under epithelial-mesenchymal transition can be easily taken.”

In the revision, there is no attempt to evaluate the effects of the threshold values. The description of the integrated analysis is not sufficient.

Answer: The determination of two groups regarding CTC1 and (CTC2-CTC1) values were performed through Lausen and Shumacher, 1992 approach. In this method all observed CTC values were evaluated in order to maximize the standardized log-rank statistic. The CTC value that maximizes the statistic is considered as the cut-off. In order to illustrate the idea graphically, in this new version of manuscript we included the plots of observed CTCs in relation to the standardized log-rank statistic. Note that CTC1 equals 1.5 and CTC2-CTC1 equals 5.5 are the values maximizer of the standardized log-rank statistic, consequently these values are the best cut-offs in order to discriminate the two curves".

No clear rationale for using circulating tumor cells for metastatic colorectal cancer. 

Answer: we hope you appreciate the paragraph we included in the Introduction section.
